# Human Raters Cannot Distinguish English Translations from Original English Texts

**Shira Wein**
Georgetown University
sw1158@georgetown.edu

## Abstract

The term translationese describes the set of linguistic features unique to translated texts, which appear regardless of translation quality. Though automatic classifiers designed to distinguish translated texts achieve high accuracy and prior work has identified common hallmarks of translationese, human accuracy of identifying translated text is understudied. In this work, we perform a human evaluation of English original/translated texts in order to explore (1) raters' ability to classify texts as being original or translated English and (2) the features that lead a rater to judge text as being translated. Ultimately, we find that, regardless of the annotators' native language or the source language of the text, annotators are unable to distinguish translations from original English texts and also have low agreement. Our results provide critical insight into work in translation studies and context for assessments of translationese classifiers.

## 1 Introduction

*Translationese* is the set of features that are unique to translated texts, even in high-quality translations (Kunilovskaya and Lapshinova-Koltunski, 2019). Prior work on translationese has characterized hallmarks of translated texts (Baker et al., 1993; Toury, 1980), for example when the translated text conforms more to the grammatical structure of the source language than the target language due to source language "shining through" (Teich, 2003).

It is clearly important to understand how translations actually differ from original texts in perception. Though automatic classifiers can identify translated versus original English texts (Baroni and Bernardini, 2006; Ilisei et al., 2010; Koppel and Ordan, 2011; Popescu, 2011; Rabinovich and Wintner, 2015) by detecting statistical patterns associated with features of translationese (section 2), work examining how humans distinguish translations is critically understudied. Because these statistical

tendencies associated with translations emerge at a large scale, even in natural-sounding translations (Kunilovskaya and Lapshinova-Koltunski, 2019), this raises the question of whether individual translated passages exhibit discernible features which distinguish them from text originally written in the language.

Building on a prior study which assessed Finnish speakers' ability to identify translated texts (Tirkkonen-Condit, 2002), in this work we perform (to our knowledge, the first) comprehensive study for English texts assessing whether human raters are able to distinguish human-produced English translations from original English texts. Specifically, we ask 8 human raters (both native and non-native speakers of English) to judge 120 English passages as being translated or originals. Our selected data spans 6 genres. We consider 5 research questions:

**RQ1** Can human raters reliably (above chance) make a binary judgment as to whether a sentence is originally written in or translated into English?

**RQ2** Do raters agree on translationese judgments (regardless of accuracy)?

**RQ3** Are native speakers able to make this judgment more accurately than nonnative speakers?

**RQ4** Does source language affect rater accuracy?

**RQ5** Does genre affect rater accuracy?

We perform empirical analyses to examine the relationship between each variable (speakers' native language, source language of the text, and genre) and human raters' ability to distinguish English translations from original English texts, thus indicating how each variable affects translation perception itself.

Where mean accuracy is denoted as $\mu_a$ and $M = 0.5$, we define our null hypothesis $\mathcal{H}_0$ as $\mu_a \leq M$ as we portend that human raters will not be able

to distinguish English translations from original English texts at a rate above chance. The alternate hypothesis $\mathcal{H}_1$ is thus defined as $\mu_a > M$.

We expect that this work will provide much-needed foundational insight into translation studies and may also be useful for engineers building translationese classifiers.

## 2 Related Work

The term *translationese* describes the set of features commonly found in translated texts that do not appear in text originally written in that language (Gellerstam, 1986), including interference from the source language or over-normalizing to the target language. Features of translationese include simplified language in the translated text (Blum and Levenston, 1978), overuse of cohesive markers and overspecification of implied information in the source text per the explicitation hypothesis (Blum-Kulka, 1986). Human and machine translations exhibit different characteristics of translationese (Bizzoni et al., 2020). Language-specific differences have been observed between original and translated texts in Chinese (Xiao, 2010) and Spanish (Ramón, 2015), among other languages.

From an engineering perspective, the presence of translationese also has a negative effect on system performance and evaluation. Translationese in test sets can lead to inflated and inaccurate evaluation scores (Graham et al., 2020) and when incorporated into training data, the performance of the model may also be affected (Ni et al., 2022; Yu et al., 2022). Recent work has set out to mitigate the amount of translationese in embeddings (Dutta Chowdhury et al., 2022) and in text itself (Wein and Schneider, 2023).

The most related work to ours is Tirkkonen-Condit (2002), a translationese study which performed two pilots asking human raters to judge Finnish texts as being translated or original (a binary judgment). The first pilot included 27 annotators and used 20 original Finnish passages and 20 translated Finnish passages, from various domains/genres. Extra-linguistic identifiers such as named entities were filtered out as to not hint to whether the text was translated or not. In the second pilot, 74 teachers of Finnish as a foreign language judged 6 of the texts from the first pilot. The study found that the accuracy of humans to distinguish translated or original Finnish texts was 61.5% for pilot one and 63.1% for pilot two, only slightly above chance. Tirkkonen-Condit (2002) also found that "unique" items—such as collocations, idioms, and language-specific features— are often an accurate indicator of an original Finnish text.

In our work, we expand on Tirkkonen-Condit's (2002) study in four notable ways. First, we perform the first human evaluation of translationese for English, shining a light on whether humans' low accuracy at distinguishing original/translated texts holds beyond Finnish. Second, while Tirkkonen-Condit (2002) seems to have only collected judgments from native speakers of Finnish, we collect annotations from both native and nonnative speakers of English. Third, we explore whether genre affects rater accuracy. Finally, we incorporate a statistical rigor into our study by addressing each research question with empirical analyses.

## 3 Methodology & Experimental Design

In order to understand how translations differ from original texts and human perception of those differences, we ask fluent speakers of English to judge passages as being either originally English or translated into English.

**Data.** We sample data from Rabinovich et al. (2018) and Tolochinsky et al. (2018), which contain a mix of human translated and original English sentences as well as metadata marking each sentence as translated/original. The data consists of European Parliament proceedings, Canadian Hansard (parliamentary proceedings), literature, political commentary, TED talks (Rabinovich et al., 2018), and United Nations proceedings (Tolochinsky et al., 2018). Before sampling from these datasets, we manually filtered out passages which clearly revealed the source language or nationality of the speaker (such as "I would therefore once more ask you to ensure that we get a Dutch channel as well," and "the Government of the Russian Federation finds it necessary to [...]").

From these datasets, we randomly select a uniform sample of 60 passages originally in English and 10 translated passages originally written in each of the following 6 languages: Arabic, Chinese (Mandarin), French, German, Russian, and Spanish. This results in an even split between original and translated English passages (120 passages total, 60 translated and 60 originally English). Following Tirkkonen-Condit (2002), which used passages of length 100 - 300 tokens, we collect passages between 90 and 300 tokens (adapted slightly to fit

| Group | Nonnative | Native |
|---|---|---|
| Mean | 0.4875 | 0.5063 |
| SD | 0.0812 | 0.0463 |
| SEM | 0.0406 | 0.0232 |

Table 1: Mean, standard deviation (SD), and standard error of mean (SEM) of accuracies for the native and nonnative speaker groups.

natural cutoff points off our data).

**Human Evaluation.** In order to investigate whether translations are more identifiable by native or nonnative speakers, we collect (binary) judgments from four native and four nonnative speakers of English. All annotators are graduate students fluent in English with background in linguistics. The annotators have not received special training in the specific task of translation detection prior to completing the task.

The instructions presented to the annotators can be seen in Figure 1. The annotators mark each passage as being **O** (originally English) or **T** (translated into English) and are able to provide additional comments justifying their annotations (discussed in Section 5).

## 4 Results & Discussion

**RQ1** *Can human raters reliably (above chance) make a binary judgment as to whether a sentence is originally or translated English?* We calculate accuracy for each annotator and with these accuracies we perform a one-sample t-test. In order to determine whether annotators are able to perform above chance ($> 0.5$), we set the hypothetical mean to 0.5 and test whether the two-tailed $p$-value indicates a statistical significance between the hypothetical mean and the actual mean of accuracies. The two-tailed $p$-value$= 0.8907$, which is not a statistically significant $p$-value, so we fail to reject the null hypothesis $\mathcal{H}_0$: $\mu_a \leq M$ where $M = 0.5$. The actual mean of accuracies for the 8 annotators is 0.4969, with a standard deviation of 0.0620. Thus, **no**, human raters are not able to perform above chance when distinguishing original/translated English.

**RQ2** *Do raters agree on translationese judgments (regardless of accuracy)?* In order to assess annotators' agreement on translationese judgments, we calculate inter-annotator agreement via Cohen's Kappa. The inter-annotator agreement across all 28 calculations (every combination of the 8 annotators) produces a mean of 0.0706 and ranges from

-0.1933 to 0.2236, which is very low agreement for even the most similar pair of annotators. Therefore, **no**, annotators do not agree on translationese judgments. Interestingly, even passages with high agreement tend to be inaccurate. Of the 6 passages which receive 6 or 7 judgments of T (none received all 8 Ts), only 3 were in fact translations. Of the 9 passages for which all raters gave a judgment of O, only 3 of the 9 passages were actually originally English. 11 passages received 7/8 O judgments and 6 of those 11 were indeed originally English.

**RQ3** *Are native speakers able to make this judgment more accurately than nonnative speakers?* **No**, native speakers are not able to more accurately distinguish translated/original English texts than nonnative speakers. Again using annotator accuracies, we perform a two-sample t-test, which results in a two-tailed $p$-value$= 0.7023$, indicating that the difference is not statistically significant. The mean, standard deviation, and standard error of means for both groups of speakers can be seen in Table 1.

**RQ4** *Does source language affect rater accuracy?* First, for each of the 7 original languages of the text (Arabic, German, English, Spanish, French, Russian, and Mandarin), we calculate accuracies for each annotator (seen in Table 2) and perform a one-way ANOVA. The $p$-value for the ANOVA equals 0.0130, indicating that at least one treatment is statistically different. Examining the results of a Tukey Honest Significant Difference (HSD) test (Gleason, 1999), we find that the English sample is statistically significantly different from the Arabic, German, and Russian samples.

Moving beyond specific language pairs, now we consider whether annotator accuracy is statistically different between the non-English source languages, as well as whether there is a statistical difference in annotator accuracy between English texts and all non-English texts (grouped together). When removing English and performing a one-way ANOVA on only the 6 non-English languages, the $p$-value$= 0.8063$, indicating that there is **no statistical difference between the 6 non-English source languages**. We then measure the statistical difference between English and all non-English languages with a two-sample t-test, and find that the two-tailed $p$-value$= 0.0008$, revealing that there is a **statistically significant difference in annotator accuracy for English and non-English**. The mean English accuracy was 0.6458 (standard deviation

Thank you for taking part in this human evaluation. In the Survey tab, you will find 120 human-produced English passages, some of which were originally uttered in English, and some of which were originally uttered in a non-English language and then translated (by a human) into English. The question we would like you to answer is:

**Does this sentence sound like it was originally written in English or translated from another language into English?**

There are two options: (1) Originally English, i.e. sentence was originally uttered in English, or (2) Translated into English, i.e. the sentence was originally uttered in a non-English language and then human translated into English. For each sentence, pick either (1) or (2) by putting an **O** or **T** to indicate O for *Originally English* and T for *Translated into English*.

For example, if you think sentence example A was produced originally in English, this is how you would mark it in the Survey tab:

| Sentence | **O**: Originally English / **T**: Translated into English |
|---|---|
| This is example sentence A. | O |

If you have any additional comments or thoughts on a sentence (e.g. if there is a specific feature of the sentence you would like to point out), you are able to optionally enter any comments into column C of the Survey Tab, marked [Optional] Comments.

Figure 1: The instructions presented to human annotators prior to classifying the passages as translated or originally English.

| Language | AR | DE | EN | ES | FR | RU | ZH |
|---|---|---|---|---|---|---|---|
| Anno 1 | 0.8000 | 0.7000 | 0.6833 | 0.4000 | 0.3000 | 0.5000 | 0.4000 |
| Anno 2 | 0.3000 | 0.6000 | 0.5000 | 0.4000 | 0.6000 | 0.3000 | 0.1000 |
| Anno 3 | 0.2000 | 0.300 | 0.7000 | 0.3000 | 0.3000 | 0 | 0.1000 |
| Anno 4 | 0.2000 | 0.2000 | 0.5167 | 0.5000 | 0.5000 | 0.3000 | 0.6000 |
| Anno 5 | 0.6000 | 0.2000 | 0.6333 | 0.7000 | 0.5000 | 0.5000 | 0.4000 |
| Anno 6 | 0.1000 | 0.3000 | 0.5000 | 0.5000 | 0.4000 | 0.4000 | 0.3000 |
| Anno 7 | 0.3000 | 0.1000 | 0.7167 | 0.5000 | 0.4000 | 0.2000 | 0.5000 |
| Anno 8 | 0 | 0 | 0.9167 | 0 | 0 | 0 | 0.3000 |

Table 2: Mean accuracy for each annotator for English (EN) and for each of the 6 source languages translated into English: Arabic (AR), German (DE), Spanish (ES), French (FR), Russian (RU), and Chinese (ZH).

of 0.1425) while the mean non-English accuracy was 0.3354 (standard deviation of 0.1508). This demonstrates that annotators are able to more accurately identify an originally English passage as being such, though this is likely due to the fact that the raters' judgments were imbalanced and marked more passages as originals than translations (an average of 78 / 120 passages across annotators were judged as originals). In particular, annotator 8 judged most of the passages to be originals, leading to high accuracy for the English passages, and very low accuracy for the non-English passages (again seen in Table 2).

**RQ5** *Does genre affect rater accuracy?* **Yes**, when performing a one-way ANOVA on the mean accuracies for each annotator of each genre, the *p*-value= 0.0272, showing that there is a statistical difference in mean accuracy by genre. Per Tukey's HSD, we find that there is a statistically significant difference between the following two treatments: Literature and Political Commentary, and Political Commentary and United Nations proceedings. This is indicative of annotators having more difficulty judging literary texts and United Nations commentary than the political commentary (see accuracies in Table 3). The literary texts are mostly from the 1800s and are artistic in nature, making them more challenging to parse as translations or originally English. The literary texts are mostly originally English, so source language (re: RQ4) likely does not serve as a confounding variable here. Similarly, the United Nations proceedings are translations of speech and contain more complex topics. On the other hand, the political commentary genre is the most well-formed and structured in nature of the genres considered, which may be why it was

| Genre | Europarl | Hansard | Literature | Politics | TED | UN |
|-------|----------|---------|-----------|----------|-----|-----|
| Anno 1 | 0.6154 | 0.7500 | 0.3750 | 0.7857 | 0.6667 | 0.5660 |
| Anno 2 | 0.6154 | 0.2500 | 0.3750 | 0.5714 | 0.4167 | 0.4340 |
| Anno 3 | 0.1000 | 0.6667 | 0.3750 | 0.7143 | 0.5833 | 0.3019 |
| Anno 4 | 0.5385 | 0.4167 | 0.4375 | 0.5714 | 0.5833 | 0.4717 |
| Anno 5 | 0.7692 | 0.2500 | 0.6250 | 0.7143 | 0.6667 | 0.4906 |
| Anno 6 | 0.5385 | 0.5000 | 0.2500 | 0.6429 | 0.5000 | 0.3396 |
| Anno 7 | 0.5385 | 0.5000 | 0.5625 | 0.6429 | 0.4167 | 0.5094 |
| Anno 8 | 0.3846 | 0.7500 | 0.5000 | 0.6429 | 0.5833 | 0.3774 |

Table 3: Mean accuracy for each annotator for each of the 6 genres ("politics" refers to political commentary).

the easiest to distinguish as translated or originally English.

## 5 Discussion of Annotator Comments

Annotators generally described the task as challenging and noted particular difficulty with passages containing technical or "bureaucratic" language.

Annotators commented on "clues" which led them to their judgments, such as framing/content ("comparing modern geopolitics to ancient Greece seems like an American/British thing to do") and date formatting. Linguistic features such as idioms (e.g."its credibility is now *shot*") and metaphorical language (e.g. "picture I am painting") led annotators to mark a passage as originally English. Appropriate use of infrequent prepositions, such as "amidst," also caused annotators to believe the passage was originally composed in English. On the other hand, annotators cited surprising preposition choice (e.g. "following *on* the earlier attacks") and the use of excessive modifiers as indicators of text being a translation. Unnatural sentence structure, unnatural sounding language (e.g. "fund of information"), under/overspecification, and repetition were all also noted in comments as reasons to annotate a passage as a translation. Intriguingly, though the annotators achieved low accuracy, most of the reasons cited as being reasons to judge a text as a translation are established hallmarks of translationese (Volansky et al., 2013).

## 6 Conclusion

In this study we assessed human raters' ability to identify English translations as distinct from original English texts and found that raters are not able to accurately make this distinction or agree with other annotators. Our statistical analyses revealed that, interestingly, human raters' (in)ability to distinguish English translations from original English text persists regardless of the speakers' native language or the source language of the text. We found

that even passages with high agreement were often inaccurately annotated. With respect to genre, formal written texts are able to be distinguished more accurately than speech or literary works.

Though existing translationese classifiers are able to separate translated and original texts with high accuracy (>90%) (Pylypenko et al., 2021), human raters are not able to do so on the same genres. This is likely because classifiers are picking up on more subtle cues that can be measured statistically but are not easily visible to human annotators; this is validated by Amponsah-Kaakyire et al. (2022), which demonstrates that hand-crafted features of translationese are only a small portion of what BERT-based translationese classifiers learn, and these hand-crafted features are even subsumed under what BERT learns without features. Our results further indicate that there are underlying properties of texts that separate translations from originals which are not easily visible in the surface form to human annotators and that, in line with prior work, well-known features of translations do not fully encompass "translationese" on a statistical scale.

## Limitations

While our empirical/statistical and qualitative analyses provide novel insight into human perceptions' of English translations, engineering implications of our findings are largely left for future work, though we do draw comparisons to translationese classifiers in Section 6. We are limited in our study by the availability of publicly available corpora which have both translated and original English passages in the same context, thus constraining the types of genres and source languages we are able to pull from for our study.

## Acknowledgements

Thanks to anonymous reviewers for their useful feedback and to Rotem Dror for her guid-

ance on significance testing. Thanks also to Tatsuya Aoyama, Aryaman Arora, Shabnam Behzad, Michael Kranzlein, Lauren Levine, Jessica Lin, Yang Liu, and Wesley Scivetti. This work is supported by a Clare Boothe Luce Scholarship.

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
