# OpenReview forum: "Human Raters Cannot Distinguish English Translations from Original English Texts"
_EMNLP/2023/Conference — EMNLP 2023 Main_

### Official Review · Reviewer_J1E6 · 2023-08-04

**Soundness:** 3

**Excitement:**

2: Mediocre: This paper makes marginal contributions (vs non-contemporaneous work), so I would rather not see it in the conference.

**Missing References:**

The first paragraph in introduction mentions a well-established evidence on translated texts, but the citation choice is somewhat weak; there are plenty of seminal studies on the differences between original and translated texts, to name a few: GideonToury, 1980. In Search of a Theory of Translation; William Frawley, 1984. Prolegomenon to a theory of translation; Mona Baker, 1993. Corpus linguistics and translation studies: Implications and applications; Martin Gellerstam, 1986. Translationese in Swedish novels translated from English.


**Paper Topic And Main Contributions:**

This short paper studies the question whether human annotators can reliably distinguish between text originally written in some language (English) or translated from other language to English (translationese). The authors put forward four research questions and show that the task is very difficult for humans, despite the considerable degree of success on the same objective with automatic classifiers.


**Questions For The Authors:**

RQ1 – did you apply any constraints on the sentence length for sentences subject to annotation? Extremely short sentences could be impossible for this task. I would even suggest to go beyond the sentence level – short paragraphs would constitute a more reliable setup.

lines 140-146: was filtering done manually or automatically? If automatically, it would be useful to provide more details on the procedure.

What were the guidelines provided to the annotators? Were they presented with any indicators or clues they should pay attention to? Were they familiar with the concept of translationese?

Regarding the clues that helped the annotators conclude about a sentence being original or translated: I do not find framing or content (line 277), or date formatting (line 280) particularly interesting, since these are not markers of translationese but rather cultural and topical indicators.


**Reasons To Accept:**

I find the work is quite interesting, generally clear and well-written.

All experiments are framed in a sound statistical manner, and supported by significance tests.


**Reasons To Reject:**

The contribution, novelty, technical depth and potential impact of this work are somewhat lightweight, even for a short paper IMO. One possible way to enhance the work is to perform a range of experiments with automatic classifiers on the same set of datasets and examine the patterns highlighted by classifiers vs. those the annotators could capture.

The research questions presented in the paper are not sufficiently motivated. The authors say that “work examining how humans distinguish translation is critically understudied”, but it is still not clear why that’s an important question: can you tie it to some cognitive aspects of language perception?


**Reproducibility:**

4: Could mostly reproduce the results, but there may be some variation because of sample variance or minor variations in their interpretation of the protocol or method.

**Reviewer Confidence:**

4: Quite sure. I tried to check the important points carefully. It's unlikely, though conceivable, that I missed something that should affect my ratings.

**Typos Grammar Style And Presentation Improvements:**

line 52: originally  originally written in English?

---

> ### Author Rebuttal · Authors · 2023-08-26
>
> Thank you for your comments. First we will answer your questions and then respond to your comments.
>
> Regarding RQ1: we do not perform annotations at the sentence level, and instead annotate short paragraphs. As we mention in the paper (line 156), following Tirkonnen-Condit (2002), which used passages of length 100 - 300 tokens, we collect passages between 90 and 300 tokens (adapted slightly to fit natural cutoff points off our data).
>
> “was filtering done manually or automatically?” Filtering was performed manually; we will clarify this in the final version.
> Regarding your question “What were the guidelines provided to the annotators?” As we note in Section 3 (line 161), we have provided the specific instructions and guidelines provided to annotators in Appendix A.
>
> For your comment “Regarding the clues that helped the annotators conclude about a sentence being original or translated: I do not find framing or content (line 277), or date formatting (line 280) particularly interesting, since these are not markers of translationese but rather cultural and topical indicators.” We agree, these are less exciting markers of translationese, though still important to note, as they were reasons cited for annotators’ decisions and they do play a role in human perception of text being translated or original. Other cited reasons such as preposition (line 286) lead to more open questions.
>
> We appreciate the suggestion for comparing human performance against automatic classifiers, though the question we are investigating here is solely in regard to human capability. Existing literature has already demonstrated that, at the statistical level (across a whole training corpus), classifier models are able to distinguish translated and original texts at a high accuracy level. Instead, in this work, we investigate whether fluent English speakers are able to classify individual passages as translated or not, and empirically investigate five research questions related to this.
>
> Regarding your comment “The authors say that “work examining how humans distinguish translation is critically understudied”, but it is still not clear why that’s an important question,” as we mention in the paper, it is important to understand how translations actually differ from original texts in perception. If we want to make progress in translation studies, as well as in effective human evaluation of translation, understanding how humans perceive translations is critical and yet notably underexplored. In the final version, given more space to flesh out the motivation, we will do so.
>
> Thank you for your reference suggestions, we will incorporate those into the final version.
>
> We hope this resolves anything you have missed or wondered about during your initial read of our paper and that this enables you to re-assess your review, in particular with regard to soundness.

---

### Official Review · Reviewer_xarF · 2023-08-05

**Soundness:** 4

**Excitement:**

4: Strong: This paper deepens the understanding of some phenomenon or lowers the barriers to an existing research direction.

**Paper Topic And Main Contributions:**

This paper discusses highly interesting findings in terms of human raters' ability to identify whether an English sentence has been natively written as such or has been translated from another language. The source sentences span several languages among which Arabic, German, English, Spanish, French, Russian, and Mandarin.
The paper concludes with rather low inter-annotator agreements therefore pointing to the fact that humans seem not to be good at actually identifying whether a sentence has been translated or not.

**Questions For The Authors:**

* Table 2 does seem to point to "outlier" annotators, i.e. people that seem to perform very well, e.g. Anno 1 in AR and DE, or Anno 8 in EN -- any explanations / findings around that which could be mentioned?

**Reasons To Accept:**

* very interesting and highly debated research question
* corroborating the results nicely, with thorough experimentation
* presented very well and clearly written
* extendable and could even inform future classifiers in the detection of translated material

**Reasons To Reject:**

* corpora used could have been attempted to separate between both translated and original English sentences in the first place -- there are corpora available like that, e.g. Europarl which gives indications whether a passage is native or not
* not much detail provided who the annotators are and how well they have (not) been trained for the task -- and whether that would make a difference at all?

**Reproducibility:**

4: Could mostly reproduce the results, but there may be some variation because of sample variance or minor variations in their interpretation of the protocol or method.

**Reviewer Confidence:**

4: Quite sure. I tried to check the important points carefully. It's unlikely, though conceivable, that I missed something that should affect my ratings.

---

> ### Author Rebuttal · Authors · 2023-08-26
>
> Hello, thank you for your thoughtful review and comments. Regarding the note that "corpora used could have been attempted to separate between both translated and original English sentences in the first place:” we agree, these distinctions certainly exist in a few corpora. We used two of these datasets for our annotation in this work. However, our research question is whether humans are able to distinguish between translated and original English sentences.
>
> Regarding “not much detail provided who the annotators are and how well they have (not) been trained for the task -- and whether that would make a difference at all?” Thank you for noting this, in the final version we will add more information on this. The annotators are all fluent English speakers and are all graduate students with some linguistic knowledge, making them even better suited to distinguish original and translated English than the average English speaker. As noted in the paper, of the 8 annotators, 4 are native and 4 are nonnative English speakers.
>
> “Table 2 does seem to point to "outlier" annotators […] any explanations / findings around that which could be mentioned?” As we mention in the paper, annotators are able to more accurately identify an originally English passage as being such, though this is likely due to the fact that the rater's judgments were imbalanced and marked more sentences as originals than translations (average of 78 / 120 passages judged as originals).  This is especially true for Annotator 8, as you note— we will add this to the paper. For other Annotators who more accurately mark the translated passages, they tend to mark a higher portion of the passages as being translated.

---

### Official Review · Reviewer_p1Sh · 2023-08-11

**Soundness:** 3

**Excitement:**

3: Ambivalent: It has merits (e.g., it reports state-of-the-art results, the idea is nice), but there are key weaknesses (e.g., it describes incremental work), and it can significantly benefit from another round of revision. However, I won't object to accepting it if my co-reviewers champion it.

**Paper Topic And Main Contributions:**

This work studies humans' ability to differentiate between text originally written in English and translated text. The findings demonstrate that regardless of the raters' native language or the source language, they are unable to make such distinctions effectively.



**Questions For The Authors:**

- I feel like somewhere earlier in the paper you need to make it clear that the translations are human-generated translation
- For this study and dataset, exploring how a classifier baseline's performance compares to that of human raters could provide insightful information
- Your discussion section ended with: “though the annotators achieved low accuracy, most of the reasons cited as being reasons to judge a text as a translation are established hallmarks of translationese.”
does not hold the same significance? Or the genre of the text might play a role here; EuroParl and Hansard texts appear more protocol-based compared to open-domain texts.







**Reasons To Accept:**

It focuses on the theoretical aspects of translation and provides valuable insights into the field of translation studies.

These insights can contribute to the establishment of more equitable and robust metrics for assessing both manual and automated translations.



**Reasons To Reject:**

The paper lacks a discussion on the motivation behind the research and what would be some practical impact of the findings.  For example:
- Why did we anticipate differences between two human-generated pieces of text (translation and original text) in the first place?
- What aspects of these results were surprising?
- How might these results reshape our perspective on machine translation research?
- Following the same analogy, It would be insightful to see how this study could extend to AI-generated text. For instance, does text generated in response to Q/A or paraphrasing tasks differ from text produced in response to translation prompts? Can a classifier or human rater distinguish between them?


**Reproducibility:**

3: Could reproduce the results with some difficulty. The settings of parameters are underspecified or subjectively determined; the training/evaluation data are not widely available.

**Reviewer Confidence:**

3: Pretty sure, but there's a chance I missed something. Although I have a good feel for this area in general, I did not carefully check the paper's details, e.g., the math, experimental design, or novelty.

---

> ### Author Rebuttal · Authors · 2023-08-26
>
> Thank you for your review. We will respond to your questions and comments.
>
> Regarding your comment “The paper lacks a discussion on the motivation behind the research and what would be some practical impact of the findings” as we mention in the paper, it is important to understand how translations actually differ from original texts in perception. If we want to make progress in translation studies, as well as in effective human evaluation of translation (such that machine translation models produce outputs most preferred by humans), understanding how humans perceive translations is critical and yet notably underexplored. In the final version, given more space to flesh out the motivation, we will do so.
>
> We will consider comparing against a classifier and/or expanding to AI-generated text in future work. Existing literature has already demonstrated that, at the statistical level (across a whole training corpus), classifier models are able to distinguish translated and original texts at a high accuracy level. Instead, in this work, we investigate whether fluent English speakers are able to classify individual passages as translated or not, and empirically investigate five research questions related to this. Extending to AI-generated text is also an interesting idea for future work, again separate from our main goals with this work. If these suggestions are impacting your review, we ask you to reconsider as these are separate research questions.
>
> Thank you for the suggestion to clarify earlier that the translations are human-generated; we will add this to the Data paragraph in Section 3 in the final version.
>
> “Your discussion section ended with: “though the annotators achieved low accuracy, most of the reasons cited as being reasons to judge a text as a translation are established hallmarks of translationese.” does not hold the same significance?” We are not sure what you mean here so unfortunately we are not able to respond to this.

---

### Meta-Review · Area_Chair_FhmT · 2023-09-17

**Recommendation:** 4

**Metareview:**

This work evaluates human abilities to identified original & translated texts.

All three reviewers agree that the soundness of the proposal is either good or strong, while there is more variability with respect to excitement (varying from 2-4).  The three reviewers have raised interesting points of strength and weakness during the review period, to which the authors have provided clear answers during the rebuttal period. Although there is some disagreement between the reviewers with respect to the issues of criticism, they are compatible and enriching.

Should the manuscript be accepted, I recommend the authors to incorporate the suggestions made by reviewers, including all the details possible on the methodology (e.g. annotators, etc.), a detailed motivation and context/scope of the work, the missing references and typos. The suggestion to carry out additional experiments is interesting for future research but, in my opinion, should not preclude publication.

---

### Decision · Program_Chairs · 2023-10-07

**Decision:**

Accept-Main

**Comment:**

This work evaluates human abilities to identified original & translated texts.

All three reviewers agree that the soundness of the proposal is either good or strong, while there is more variability with respect to excitement (varying from 2-4).  The three reviewers have raised interesting points of strength and weakness during the review period, to which the authors have provided clear answers during the rebuttal period. Although there is some disagreement between the reviewers with respect to the issues of criticism, they are compatible and enriching.

Should the manuscript be accepted, I recommend the authors to incorporate the suggestions made by reviewers, including all the details possible on the methodology (e.g. annotators, etc.), a detailed motivation and context/scope of the work, the missing references and typos. The suggestion to carry out additional experiments is interesting for future research but, in my opinion, should not preclude publication.